# Antimicrobial use on Italian Pig Farms and its Relationship with Husbandry Practices

**DOI:** 10.3390/ani10030417

**Published:** 2020-03-02

**Authors:** Jacopo Tarakdjian, Katia Capello, Dario Pasqualin, Andrea Santini, Giovanni Cunial, Annalisa Scollo, Alessandro Mannelli, Paola Tomao, Nicoletta Vonesch, Guido Di Martino

**Affiliations:** 1Istituto Zooprofilattico Sperimentale delle Venezie, Viale dell’Università 10, 35020 Legnaro, Padova, Italy; kcapello@izsvenezie.it (K.C.); dpasqualin@izsvenezie.it (D.P.); asantini@izsvenezie.it (A.S.); gcunial@izsvenezie.it (G.C.); gdimartino@izsvenezie.it (G.D.M.); 2Swivet Research, 42123 Reggio Emilia, Italy; scollo@suivet.it; 3Department of Veterinary Sciences, University of Torino, 10124 Torino, Italy; alessandro.mannelli@unito.it; 4Department of Occupational and Environmental Medicine, Epidemiology and Hygiene, Italian Workers’ Compensation Authority (INAIL), Monte Porzio Catone, 00078 Rome, Italy; p.tomao@inail.it (P.T.); n.vonesch@inail.it (N.V.)

**Keywords:** AMU, swine, DDD, CIA, tail biting, management

## Abstract

**Simple Summary:**

Quantifying antimicrobial use (AMU) in livestock using dose-based methods is useful for identifying potential risk factors related to AMU and to promote responsible AMU. Herewith, we investigated the temporal patterns and effects of several structural and management factors on AMU in a sample of pig farms in northern Italy during a period of three years. Overall, AMU showed a large variability between farms and no significantly decreasing trend. However, in several farms, a significant AMU reduction of up to two-thirds was observed. Farm size, number of farm workers, air quality, average pig mortality, and presence of pigs with undocked tails on the farm had no significant effect on AMU, whereas higher welfare standards were significantly associated with lower AMU.

**Abstract:**

The analysis of antimicrobial use (AMU) data in livestock allows for the identification of risk factors for AMU, thereby favoring the application of responsible AMU policies on-farm. Herewith, AMU in 36 finishing pig farms in northern Italy from 2015–2017 was expressed as defined daily doses for Italian pigs (DDDita) per population correction unit (DDDita/100kg). A retrospective analysis was then conducted to determine the effects of several husbandry practices on AMU. Overall, AMU ranged between 12 DDDita/100kg in 2015 and 8 DDDita/100kg in 2017, showing no significant trends, due to the large variability in AMU between farms. However, a 66% AMU reduction was observed in 19 farms during 2015-2017. Farm size, number of farm workers, air quality, average pig mortality, and presence of undocked pigs on the farm had no significant effects on AMU. Rather, welfare-friendly farms had 38% lower AMU levels (*p* < 0.05). In conclusion, animal welfare management seems to be relatively more important than farm structure and other managerial characteristics as drivers of AMU in finishing pig farms.

## 1. Introduction

Antimicrobial resistance (AMR) is a global threat for both human and animal health, as the loss of antimicrobial efficacy undermines the outcome of antimicrobial treatments [1]. Several factors, including antimicrobial misuse and overuse, contribute to selection of resistant bacteria [2]. Indeed, bacteria exposed to sub-therapeutic concentrations of antimicrobials are more likely to develop resistance as an adaptive response to such exposure [3]. Livestock fed with medicated feed is likely to receive suboptimal antimicrobial treatments, as drugs are administered as mg of active substance per kg of feed. Moreover, animals suffering from loss of appetite may ingest smaller amounts of feed and consequently receive lower doses of antimicrobials [4].

According to the 2017 Joint Inter-agency Antimicrobial Consumption and Resistance Analysis (JIACRA) report on antimicrobial use (AMU) in food producing-animals in Europe, Italy ranked third after Cyprus and Spain [5]. To promote responsible AMU, the World Health Organization (WHO) has grouped antimicrobials into different categories based on their importance in human medicine: critically important antimicrobials (CIA) (further divided into Highest Priority CIAs and High Priority CIAs), highly important antimicrobials, and important antimicrobials. This distinction was made based on the availability of alternative therapies and the frequency of use of active substances in humans [6]. More recently, the European Medicines Authority (EMA) made another classification by grouping antimicrobials in four classes according to the potential consequences to public health when used in animals: A (avoid—not authorized in veterinary medicine), B (restrict), C (caution), and D (prudence) [7]. 

Previous studies have found some herd-specific interventions concerning biosecurity and health management to be associated with AMU reduction in different European countries [8]. Conversely, rearing undocked pigs might lead to increased AMU, due to tail biting related sequelae [9]. Indeed, tail biting is a major issue in pig production due to the risks of secondary infections, reduced performance, increased mortality, and the condemnation of carcasses [10]. However, it has recently been reported that some northern European countries, such as Sweden, are able to raise undocked pigs and yet maintain low AMU levels [11]

In 2017, pig production in Europe increased by 1.6% with a total number of 150.2 million animals, 8.6 million of which reared in Italy [12]. Italy’s pig farming is mostly devoted to the production of Protected Designation by Origin (PDO) dry-cured raw ham, with animals reaching at least 160 kg of live weight at nine months of age. At present, published AMU data in the pig production sector are scarce, and the need for such data is particularly important in countries like Italy where overall AMU in livestock is relatively high. The aim of the present study is to provide updated data on AMU in commercial pig farms in Italy and to identify significant associations between AMU and husbandry practices, including tail docking. AMU was quantified using dose-based methods [13], taking into account their importance for human medicine.

## 2. Materials and Methods 

### 2.1. Data Collection 

Thirty-six fattening pig farms were randomly selected from the National Farm Registry, including only those holdings with at least 900 fattening units located in northern Italy (Emilia-Romagna, Lombardy, and Veneto regions), in which 37% of the national pig population (consisting of around 7400 holdings in total) is concentrated. Farm size, in term of fattening units, ranged from 900 to 10,000, and can be considered representative of the Italian pig sector. Farms were visited once in 2017 by a veterinarian to collect the following data: number of slaughtered pigs per year, administered antimicrobial treatments, number of farm workers, average mortality rate (calculated yearly based on the introduced and slaughtered animals), presence of outdoor defecation areas, type of farm (farms working for a welfare-friendly label vs. conventional farms), presence of undocked animals (all animals, or some groups). Based on the data distribution, farms were divided into two groups according to having more or less than 2000 animals per fattening cycle (median value). During each visit, the veterinarians measured the level of ammonia in different parts of the fattening units using a DRAGER X-am 7000 (Dräger Safety AG & Co. KGaA, Lübeck, Germany), according to the sampling scheme and the threshold of acceptability proposed by the Italian Reference Center for Animal Welfare [14].

### 2.2. Data Analysis

In order to quantify the overall AMU per year, the defined daily dose/population correction unit (DDDvet/PCU) method proposed by the EMA was adopted. For each active substance, EMA provides standardized daily dosages, calculated as the average concentration of various marketed products sold in the European Union (EU) [12]. PCU was obtained by multiplying the total number of yearly slaughtered pigs by a standard weight at treatments of 65 kg, as proposed by the EMA. As the DDDvet for long acting injectable products, namely, gamithromycin, tildipirosin and tulathromycin, have not yet been published, we set up the DDD based on the SPCs of veterinary medicinal products containing those active substances.
DDDvet/PCU = total administered active substance (mg)/defined daily dosage (mg/kg/d) × n animals × expected weight at treatment (kg)(1)

To provide a higher level of precision, quantitative data on AMU were also converted into Italian defined daily doses (DDDita), i.e., taking the individual daily dosage given for each marketed product from the summary of product characteristics (SPC) [15]. When the daily dosage was provided as a range, the maximum value was chosen [16]. The number of DDDita was divided by the number of slaughtered animals produced for each of the herds enrolled in the study and retrieved in the national swine registry [17] by the expected weight at treatment, which for Italian heavy pigs was proposed to be 100 kg of live weight, in agreement with previous studies [18]. Antimicrobials were grouped according to EMA and WHO classifications [6,7].

Considering the sample size and the non-normal distribution of DDDita data, non-parametric tests were used for the statistical analysis. The Friedman test was applied to assess differences in DDDita over the years. Moreover, to assess possible associations between AMU and husbandry practices, the median DDDita values per farm were used. The Mann-Whitney two-sample statistic was then applied to test the statistical significance of these associations after having checked the equality of variances by means of the Leven’s robust test statistic. For the factors with more than two groups, the Kruskal-Wallis test was used. Data analysis was performed using STATA v. 12.1.3 software. 

## 3. Results

The total amount of AMU in the 36 finishing farms in 2015, 2016, and 2017, calculated as DDDvet/PCU, was 24.46, 18.76, and 15.91, respectively. Total and median values calculated as DDDita/100kg are given in Table 1 and showed no significantly decreasing trends. However, in 19 farms, AMU reduction was up to 66% from 2015 to 2017. Antimicrobials belonging to the B category, “Restrict”, ranged between 1.6% and 5.7% of the total AMU during the three years of study. Tetracyclines, lincosamides, and penicillins were the most frequently administered antimicrobial classes, ranging between 63% and 70% of the total AMU during the three years of study. As shown in Table 2, farm size, number of involved personnel, air quality, average pig mortality, and presence of undocked pigs did not exhibit any significant association with AMU. Rather, farms engaged in welfare-friendly pig production systems were characterized by a 38% lower AMU as compared to conventional farms (*p* < 0.05). 

## 4. Discussion

To our knowledge, this is the first study assessing AMU in Italian pig farms according to dose-based methods and their relationship with husbandry practices. This approach allowed for adding different active substances together to make a sum [13]. AMU data exhibited a high variability between farms, so that a trend of reduction could not be detected. Some farms, however, were able to decrease AMU by up to 66%. The same approach was previously adopted on AMU data from 30 Italian farrow-to-finish and fattening pig farms [18]. The authors found that pigs in the finishing phase had around twice as high AMU values as in the present study, with a decreasing trend in AMU from 2016 to 2017 (16.3% for total AMU and 45.8% for Highest Priority CIAs), which was not observed in the present study. The reported percentage of Highest Priority CIAs in 2017 was slightly higher (11% of total AMU) to the one found in the present study (i.e., 9% in 2017).

From 2015 to 2017, penicillins, tetracyclines, and lincosamides accounted respectively for 22–31%, 19–24%, and 19–24% of the total mass of administered antimicrobials in the sampled farms, representing the most frequently used drug classes. Interestingly, the European Surveillance of Veterinary Antimicrobial Consumption (ESVAC) report on sales of veterinary antimicrobials agents in the EU in 2017 shows that penicillins and tetracyclines were the most frequently sold classes of antimicrobials per PCU in Italy [15], which is in line with our data. These substances are often used to treat respiratory diseases in fattening pigs [19,20], while lincosamides are frequently used in the treatment of common infections in Italy such as *Brachyspira hyodysenteriae* [18], *Mycoplasma hyopneumoniae* [21], and *Lawsonia intracellularis* [22] infections.

At present, only few studies have studied risk factors for AMU in livestock [8,9,10,11,12,13,14,15,16]. In the poultry sector, which compared to the pig sector is a vertically integrated system, a decreasing trend in AMU was observed from 2015–2018 [15]. Moreover, several correlates of AMU have been identified, such as season, geographical area, and production type [16]. Acknowledging such significant associations in other species/production systems can be challenging, since several factors need to be taken into account. Moreover, prescribing habits of AMU might not be necessarily rational [20]. In the fattening pig sector, antimicrobials are mass-administered mainly via medicated feed or drinking water, while individual treatments are far less frequent [19]. A strategy aimed at reducing AMU should cover both veterinarians’ training and farmers’ awareness. Indeed, nowadays farmers believe that medicine consumption in humans is responsible for the increased AMR levels in animals and humans rather than veterinary AMU for therapeutic purposes [23]. Improving farm management and biosecurity measures is highly recommended, as it has shown an indirect AMU-reducing effect [24] without heavily impacting on the economic performance of the farm [8]. The proactive role of farmers is, however, of utmost importance. In a survey of Italian poultry and rabbit farmers, most of the participants believed that veterinary AMU could be decreased, with 20–30% being the most frequently chosen reduction target ranges [23]. 

Farm size (threshold level set at 2000 head per fattening cycle) had no significant effect on AMU. In the literature, conflicting findings concerning the effect of farm size on AMU are reported. Vieira and colleagues found that farms with 200–1000 slaughtered animals/year tended to use more antimicrobials than larger ones, probably because of poorer hygienic conditions compared to better organized, larger farms [25]. Inversely, other authors pointed out that larger herds (number of animals not provided) are associated with a higher AMU, explained by the increased odds of disease spread [26]. Yet, a Japanese study involving 72 pig farms found no significant differences in AMU between small and large sized farrow-to-finish farms [27].

As expected, farms applying specific welfare-friendly production systems had lower AMU levels. Indeed, this type of production is characterized by veterinary support to farmers at least every fortnight, as well as education programs for farm personnel consisting of yearly courses on good management practices, biosecurity, animal welfare, and drug usage. Moreover, routine monitoring and sanitization of drinking water pipelines after each fattening cycle are applied, and rooting material has to be assured at all times in each pen, i.e., straw (in a rack on the wall), wood stumps, and metal chains. Specific practices to avoid amputations are also in place, i.e., some or all groups with undocked tails are present and castration is permitted only under analgesia and anesthesia. 

Part of the recruited farms in this study housed undocked heavy pigs, but the presence of these pigs had no significant effect on AMU. Tail docking is a surgical intervention allowed by Dir. 2008/120EC within the seventh day of life to prevent tail biting, but not on a routine basis [28]. Italy is currently increasing the number of farms housing undocked pigs by means of a national plan [29]. One possible criticism of rearing undocked pigs is the possible increase of tail lesions in frequency and severity [30], which would require more AMU. The lack of a significant association in the present study may support the feasibility of rearing undocked heavy pigs, as previously reported in Italian studies [31,32].

High air concentrations of ammonia may severely affect the upper respiratory tract in pigs [33]. Therefore, respiratory diseases can be associated with poor air quality and, consequently, with higher AMU. Conversely, some studies highlighted a minor role of ammonia and pollutant in pigs’ gross lesions, as well as pathological changes [34], supporting the lack of a significant association found here. However, limitations due to the sample size, sampling protocol, categorization schemes, and relatively short study period cannot be excluded.

## 5. Conclusions

This study provided dose-based AMU data in a relatively small sample of Italian finishing pig farms, evidencing a high variability in AMU levels among farms. Although AMU was slightly lower than previously reported, use of antimicrobials belonging to the B category did not show a significant decrease. While welfare-friendly production systems were significantly associated with lower AMU levels, other farm structures and management characteristics, including the presence of pigs with undocked tails, did not appear to be significant drivers of AMU levels in Italian finishing pig farms.

## Figures and Tables

**Table 1 animals-10-00417-t001:** Defined daily doses per animal (total and median DDDita/100kg with 25th–75th percentile) administered in 2015–2017 in 36 Italian finishing farms.

Antimicrobial Class		2015			2016			2017	
	Total	Median	25th–75th	Total	Median	25th–75th	Total	Median	25th–75th
**Category B—Restrict**									
quinolones *	0.102	0.00	0.00–0.05	0.372	0.00	0.00–0.12	0.256	0.00	0.00–0.12
fluoroquinolones *	0.027	0.00	0.00–0.01	0.034	0.00	0.00–0.06	0.054	0.00	0.00–0.04
polymyxins *	0.058	0.00	0.00–0.00	0.101	0.00	0.00–0.07	0.010	0.00	0.00–0.00
cephalosporins *	0.005	0.00	0.00–0.00	0.002	0.00	0.00–0.00	0.000	0.00	0.00–0.00
**Category C—Caution**									
aminoglycosides	0.136	0.00	0.00–0.00	0.153	0.00	0.00–0.00	0.049	0.00	0.00–0.00
amphenicols	0.193	0.12	0.01–0.31	0.308	0.07	0.03–0.26	0.278	0.09	0.01–0.28
lincosamides	2.836	0.58	0.00–5.21	2.163	0.28	0.01–3.23	1.553	0.14	0.00–1.70
macrolides *	0.299	0.01	0.00–0.20	0.276	0.00	0.00–0.12	0.335	0.00	0.00–0.20
pleuromutilins	2.433	0.00	0.00–1.04	0.895	0.01	0.00–0.90	0.743	0.00	0.00–0.02
penicillins	2.618	1.21	0.27–3.19	2.800	2.41	0.21–5.20	1.949	1.23	0.18–4.87
aminocyclitols	0.042	0.00	0.00–0.00	0.038	0.00	0.00–0.02	0.044	0.00	0.00–0.02
**Category D—Prudence**									
tetracyclines	2.897	0.30	0.00–3.75	1.710	0.02	0.00–2.74	1.663	0.22	0.00–3.91
sulphonamides	0.247	0.00	0.00–0.00	0.063	0.00	0.00–0.00	0.345	0.00	0.00–0.00
trimethoprim	0.160	0.00	0.00–0.00	0.063	0.00	0.00–0.00	0.948	0.00	0.00–0.00
**Total DDDita/100kg**	12.051	6.24	3.73–15.97	8.975	7.16	2.93–15.84	8.226	7.57	1.22–16.09

Antimicrobials have been grouped and classified according to EMA categorization. Asterisks indicate Highest Priority-CIAs according to WHO categorization [6].

**Table 2 animals-10-00417-t002:** Associations between defined daily doses per animal (DDDita/100kg) and different variables in 2015–2017 (median value given with 25th–75th percentile) in 36 finishing farms in northern Italy.

Farm Size	N	DDDita/100kg		
		Median	25th–75th	*P*
Less than 2000 fattening units	15	7.45	5.13–23.11	0.231
More than 2000 fattening units	21	5.86	2.81–12.06	
**Type of pig industry**				
Welfare friendly labels	11	4.52	1.03–6.41	0.045
Conventional farms	21	7.30	5.00–17.43	
**Mortality**				
<2%	8	5.00	0.89–8.26	NA
2–5%	26	7.45	4.53–18.15	
>5%	2	5.49	3.20–7.77	
**Involved personnel**				
1 stockperson for more than 4000 pigs	6	5.56	0.75–12.06	0.156
1 stockperson for 2000–4000 pigs	12	5.02	3.21–7.09	
1 stockperson for less than 2000 pigs	18	7.72	5.84–20.90	
**Outdoor defecation area**				
Absence	11	5.87	3.22–29.93	0.809
Presence	21	5.94	3.56–8.93	
**Level of ammonia**				
More than 20 ppm	10	5.47	0.75–20.90	0.649
Less than 20 ppm	21	6.14	4.70–15.11	
**Presence of undocked animals**				
No	8	6.79	0.82–16.96	0.601
Some groups	9	6.76	3.58–9.06	
All animals	19	6.41	4.53–22.80	

N: number of involved farms. NA: not applicable. Significance for *p* < 0.05.

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
