# Peer review of "Antimicrobial use on Italian Pig Farms and its Relationship with Husbandry Practices"

_animals, 2020, doi:10.3390/ani10030417_

Round 1

Reviewer 1 Report

Dear authors,

thank you for sending this manuscript on AMU in Italian pig production. I was waiting for a paper like this because the pig production in Italy differs a lot from other European countries.

In my eyes the manuscript has 2 obvious weaknesses. First, material section is missing a lot of information and second, the presentation of the results leaves a lot of questions open.

Materials: Please describe your efforts to have a representative sample.

Please describe the time frame that was covered to collect data on antimicrobial treatments. Also add the number of inspections that have taken place in each farm. Was that evenly distributed over all 3 years?

Please describe the conditions of the welfare-labels? Was it the same label in each farm or just any label? Which conditions did the farmers have to fulfill? Was that linked to reduced use of antimicrobials (as is mostly the case)?

Please describe why you categorised the farm size at 2000 animals? Was that the median?

DDDita: Did you use average doses for each substance or did you take the individual doses given in each specific SPC?

Mann-Whitney-U-test only works for 2 groups. Which test did you use for the 3-group-variable "involved personnel"?

Levene's test is not necessary for non-parametric tests.

After proving differences between years, why did you decide to take the mean over years? That is a contradiction in my eyes. I would have been better to select one year.

I would prefer to read the 25th and 75th percentile instead of the IQR.

discussion: please add the reference to ll. 219 - 222. Should be [16].

Author Response

Dear Reviewer,

attached you can find the manuscript revised accordingly with your comments. You can check the amendments by using the "track changes" function in Microsoft Word. 

Major changes are highlighted in red and in yellow. Indeed we provided 25th and 75th percentiles tables and rearranged the discussion.

Below we also provided a point-by-point response. The numbers of the lines provided refer to the "no markup" version of the manuscript.

We thank you for your comments; we found them very helpful and rational in order to improve the quality of our work.

Yours,

Jacopo Tarakdjian

REPORT Reviewer #1

Comments and Suggestions for Authors

Dear authors,
thank you for sending this manuscript on AMU in Italian pig production. I was waiting for a paper like this because the pig production in Italy differs a lot from other European countries.
In my eyes the manuscript has 2 obvious weaknesses. First, material section is missing a lot of information and second, the presentation of the results leaves a lot of questions open.
Authors: thanks for the nice comment, we have included the missing details and clarified the questions left open.

Materials: Please describe your efforts to have a representative sample.

Authors: we have added more detailed information on farms recruitment and on the total population.

Please describe the time frame that was covered to collect data on antimicrobial treatments. Also add the number of inspections that have taken place in each farm. Was that evenly distributed over all 3 years?

Authors: we have clarified in the text that the collection from the registries took place in 2017 during the inspection for other parameters

Please describe the conditions of the welfare-labels? Was it the same label in each farm or just any label? Which conditions did the farmers have to fulfill? Was that linked to reduced use of antimicrobials (as is mostly the case)?

Authors: we have clarified it was the same label, with guidelines now detailed in the discussion session. There was no direct action specifically focused on AMU, but a list of actions which are likely to have an indirect impact on its need. Rows 239-246

Please describe why you categorised the farm size at 2000 animals? Was that the median?
Authors: exactly, in absence of any other reference we have chosen the median to better read the results. Row 91

DDDita: Did you use average doses for each substance or did you take the individual doses given in each specific SPC?
Authors: for DDDita, we did take individual doses given in each specific SPC. When in the SPC there was an indication of a range (e.g. 5-10 mg/kg) we did take the maximum allowed doses (i.e. 10 mg/kg). This choice was made as veterinarians reported us to be used to apply the maximum dose. Rows 107-114

Mann-Whitney-U-test only works for 2 groups. Which test did you use for the 3-group-variable "involved personnel"?
Authors: we are sorry for the oversight. For variables with three groups, the Kruskal-Wallis test was adopted; the M&M section has been modified accordingly. Rows 120-121

Levene's test is not necessary for non-parametric tests.
Authors:  the prerequisite for the applicability of both Mann-Whitney-U-test and Kruskal-Wallis test is that the two or more groups have the same shape, and so no differences in variances; the Levene’s test is appropriate to test this assumption for non-normal distribution
.

After proving differences between years, why did you decide to take the mean over years? That is a contradiction in my eyes. I would have been better to select one year.
Authors: we did not find difference between years, probably due to a high variability. This is why we have chosen to test the association between factors and median AMU, which is the most reliable parameter.

I would prefer to read the 25th and 75th percentile instead of the IQR.
Authors: addressed accordingly

Discussion
: please add the reference to ll. 219 - 222. Should be [16].
Authors: addressed accordingly. Row 200

Reviewer 2 Report

General comments

With the upcoming mandatory reporting of antimicrobial usage data by animal species for EU members, the presented data could provide important data for further work on prudent use of antimicrobials in the pig sector. However, there are some major issues concerning the manuscript. Firstly, the quality of the written English is not of standards suitable for publication. Therefore, the manuscript needs to be revised by a fluent English speaker. This could perhaps explain why the presented material is rather difficult to follow but I also believe that the manuscript must be more clearly structured.

Moreover, parts of the material and methods section are not explained in enough detail. Most importantly, the description on how the DDDita were calculated are insufficient. The methodology is described very briefly and referred to other publications in which DDDita for broilers, turkeys and rabbits have been calculated and used. However, I cannot find any DDDita for pigs. Perhaps this is described in the paper by Scoppetta but it is in Italian which makes it impossible to read for all who are not Italian speakers. I suggest that the description on how DDDita wer developed is expanded and that actual DDDita for pigs is shown, either in a table or as supplementary material. The terminology for the DDDs is also confusing as you describe both DDDita and NDDD. This must be clarified. As you use DDDita, it is difficult to get an impression on how AMU for these Italian farms relate to use in other farms, production types and countries. You need to show data in such a way that they can be compared to other AMU data. Regarding the calculations of the DDDs, I do not believe that the term PCU should be used as a different weight at the most likely time of treatment is used. Even though you describe that a different weight is used, it gives a false impression of being comparable with the ESVAC PCU. Either you decide to use PCU and then use the same weight as ESVAC or you could show it by kg. Furthermore, there is not enough information on the welfare friendly systems and how they differ to the conventional systems. This information is important as you relate AMU to production systems.

You talk about variability between farms, but this is not shown in the results. In table 1, it is not clear if median or average values are shown and there is no information on the range. I also believe that showing two decimals is not appropriate. The resolution of the data is probably not that detailed. The results on AMU are shown by WHO’s categorization of antimicrobials but recently, EMA published a categorization of antibiotics used in veterinary medicine. In my opinion, it would be more appropriate to show the results according to EMA’s categorization. 

I also think that the title is misleading as in a retrospective study as yours, you would not know for sure if it is the actual husbandry practices or technical parameters that result in a certain level of AMU. It could be the case that some practices are linked to certain parameters which could in turn give false associations to AMU. Did you consider using multivariate analysis for the statistical analysis?

I believe that the discussion could be condensed by structuring it differently. Try to be concise and discuss one finding at a time. An idea could be to begin with describing AMU patterns and then discuss possible factors impacting on AMU.

I also think that you are over-interpreting your results as only a small sub-set of farms were studied which means that you cannot draw any conclusions covering the Italian pig sector. As you have not shown the variability, you cannot draw any conclusions on this either. Neither should you include references in your conclusion.  

Specific comments

Introduction

Row 48: I do not understand “free-choice medicated feed”. This could perhaps be sorted if the text is revised by a fluent English speaker.

Row 52: I believe the JIACRA report is from 2017.

Rows 54-59: You should use the categorization recently issued by EMA. See: https://www.ema.europa.eu/en/news/categorisation-antibiotics-used-animals-promotes-responsible-use-protect-public-animal-health

Rows 59-60: This sentence is completely out of place.

Rows 62-66: this is not completely true as the Nordic countries, i.e. Norway, Finland and Sweden, all manage to raise undock tails combined with low AMU. I think that there is a lot of in formation to be found from the FareWellDock project, see. https://farewelldock.eu/

Materials and Methods

Rows 80-81: How were the farms randomly selected? This is not sufficiently described. What is implied with inspected? Visited?

Row 82: Is this the number of pigs sent for slaughter per year or fattening places?

Row 87: What is meant by threshold?

Rows 92-94: This is perhaps a matter of language, but I do not understand this, and I do not agree that you need the number of administered doses in order to be able to perform the calculations. You only need the administered amount of active substances.

Row 100: I do not understand what you mean by maximum suitable concentration.

Results

Row 115: It is not clear that medians are used in table 1. Ranges must be given.

Rows 117-118: This is not supported by the data shown.  

Row 122. I do not think that effect is the correct word. In a retrospective study you can only describe associations.

Table 2: I wonder if the data is that exact so that you can show two decimals.

Discussion

Rows 199-200: I do not understand what you mean by strictly integrated.

Rows 207-209: I do not understand this sentence.

Rows 217-219: Neither do I understand this sentence.

 Row 232: Check the spelling of Brachyspira hyodysenteriae

Row 238: I do not think that justified is the appropriate word. Perhaps explain would fit better.

Row 243-247: to be able to discuss along these lines you need to provide more detailed information on the differences between Italian welfare friendly and conventional systems. The situation may be completely different in other countries.

Rows 249-250: I think that the work performed to increase the number of farms rearing undocked pigs is commendable as strictly speaking, routine docking is not legal within the EU although in most, but not all countries, this is routine practice.

Conclusions

You cannot draw conclusion for the entire Italian pig sector based on the results for only 36 herds.

Author Response

Dear Reviewer,

attached you can find the manuscript revised accordingly with your comments. You can check the amendments by using the "track changes" function in Microsoft Word. 

Major changes are highlighted in red and in yellow. Indeed we provided 25th and 75th percentiles tables and rearranged the discussion.

Below we also provided a point-by-point response. The numbers of the lines provided refer to the "no markup" version of the manuscript.

We thank you for your comments; we found them very helpful and rational in order to improve the quality of our work.

Yours,

Jacopo Tarakdjian

REPORT Reviewer #2

General comments

With the upcoming mandatory reporting of antimicrobial usage data by animal species for EU members, the presented data could provide important data for further work on prudent use of antimicrobials in the pig sector. However, there are some major issues concerning the manuscript. Firstly, the quality of the written English is not of standards suitable for publication. Therefore, the manuscript needs to be revised by a fluent English speaker. This could perhaps explain why the presented material is rather difficult to follow but I also believe that the manuscript must be more clearly structured.
Authors: we thank the reviewer for the appreciation of our study for promoting prudent use of antimicrobials. The written English has been revised by a native English editor as requested.

Moreover, parts of the material and methods section are not explained in enough detail. Most importantly, the description on how the DDDita were calculated are insufficient. The methodology is described very briefly and referred to other publications in which DDDita for broilers, turkeys and rabbits have been calculated and used. However, I cannot find any DDDita for pigs. Perhaps this is described in the paper by Scoppetta but it is in Italian which makes it impossible to read for all who are not Italian speakers. I suggest that the description on how DDDita wer developed is expanded and that actual DDDita for pigs is shown, either in a table or as supplementary material. The terminology for the DDDs is also confusing as you describe both DDDita and NDDD. This must be clarified. As you use DDDita, it is difficult to get an impression on how AMU for these Italian farms relate to use in other farms, production types and countries. You need to show data in such a way that they can be compared to other AMU data.

Regarding the calculations of the DDDs, I do not believe that the term PCU should be used as a different weight at the most likely time of treatment is used. Even though you describe that a different weight is used, it gives a false impression of being comparable with the ESVAC PCU. Either you decide to use PCU and then use the same weight as ESVAC or you could show it by kg. Furthermore, there is not enough information on the welfare friendly systems and how they differ to the conventional systems. This information is important as you relate AMU to production systems.

Authors: we have clarified the ambiguity between DDDita and NDDD. Regarding DDDita, we have better clarified in the text that no “DDDita for pigs” exists. DDDita is a general approach of dividing milligrams of active principle (for any drug for any species) for the daily dose indicated in the summary of product characteristics with specific reference for the animal species (e.g. broiler vs turkey, etc.) or category (e.g. sows vs piglets, etc.). The only difference between DDDita and DDDvet is that DDDvet is calculated using an average daily dosage for different commercial products in EU, while DDDita is calculated on the basis of the specific daily dosage for the specific commercial product used.
To make a comparison between countries possible, we expressed the value DDDvet/PCU (using ESVAC standard weight at treatment for uniformity with other countries) together with DDDita/100kg (thereby avoiding the misleading use of the term PCU and enabling a comparison with other Italian studies).

You talk about variability between farms, but this is not shown in the results. In table 1, it is not clear if median or average values are shown and there is no information on the range. I also believe that showing two decimals is not appropriate. The resolution of the data is probably not that detailed.
Authors: we amended table 1 as requested.

The results on AMU are shown by WHO’s categorization of antimicrobials but recently, EMA published a categorization of antibiotics used in veterinary medicine. In my opinion, it would be more appropriate to show the results according to EMA’s categorization.
Authors: addressed accordingly. However, the definition of Highest priority CIAs was kept.

I also think that the title is misleading as in a retrospective study as yours, you would not know for sure if it is the actual husbandry practices or technical parameters that result in a certain level of AMU. It could be the case that some practices are linked to certain parameters which could in turn give false associations to AMU. Did you consider using multivariate analysis for the statistical analysis?
Authors: given the results of the bivariate analyses with only one factor significantly associated with AMU it was not necessary to make a multivariate analysis. Moreover, the small sample size would make difficult to assess possible interactions between factors. We understand, however, that the title can be misleading, so we modified it to:  Antimicrobial use in Italian pig farms and its relation with husbandry practices

I believe that the discussion could be condensed by structuring it differently. Try to be concise and discuss one finding at a time. An idea could be to begin with describing AMU patterns and then discuss possible factors impacting on AMU.

I also think that you are over-interpreting your results as only a small sub-set of farms were studied which means that you cannot draw any conclusions covering the Italian pig sector. As you have not shown the variability, you cannot draw any conclusions on this either. Neither should you include references in your conclusion. 
Authors: we have reduced and rearranged the discussion by putting the AMU pattern first.

Specific comments

Introduction

Row 48: I do not understand “free-choice medicated feed”. This could perhaps be sorted if the text is revised by a fluent English speaker.
Authors: addressed accordingly

Row 52: I believe the JIACRA report is from 2017.
Authors: addressed accordingly

Rows 54-59: You should use the categorization recently issued by EMA. See: https://www.ema.europa.eu/en/news/categorisation-antibiotics-used-animals-promotes-responsible-use-protect-public-animal-health
Authors: reference added

Rows 59-60: This sentence is completely out of place.
Authors: we have removed the sentence.

Rows 62-66: this is not completely true as the Nordic countries, i.e. Norway, Finland and Sweden, all manage to raise undock tails combined with low AMU. I think that there is a lot of in formation to be found from the FareWellDock project, see. https://farewelldock.eu/
Authors: we added the information as suggested.

Materials and Methods

Rows 80-81: How were the farms randomly selected? This is not sufficiently described. What is implied with inspected? Visited?
Authors: we have better detailed the approach. Rows 81-92.

Row 82: Is this the number of pigs sent for slaughter per year or fattening places?
Authors: fattening places, clarified in the text
.

Row 87: What is meant by threshold?
Authors: we have rephrased the sentence. We meant the median value used to divide farms into two groups. Row 91.

Rows 92-94: This is perhaps a matter of language, but I do not understand this, and I do not agree that you need the number of administered doses in order to be able to perform the calculations. You only need the administered amount of active substances.
Authors: we have removed the sentence and better explained the concept. We also added the equation used to calculate DDDvalues/PCU. Rows 96-114

Row 100: I do not understand what you mean by maximum suitable concentration.
Authors: we replaced “concentration” with “value”. If the daily dosage reported in the SPCs is provided as a range we choose the higher (maximum) value.

Results

Row 115: It is not clear that medians are used in table 1. Ranges must be given.
Authors: we provided the table with medians and percentiles.

Rows 117-118: This is not supported by the data shown. 
Authors: we provided the table with medians and percentiles making it consistent with the sentence.

Row 122. I do not think that effect is the correct word. In a retrospective study you can only describe associations.
Authors: we have replaced the word (row 132) and rephrased the title as:  Antimicrobial use in Italian pig farms and its relation with husbandry practices

Table 2: I wonder if the data is that exact so that you can show two decimals.
Authors: we have chosen rounding at 2 decimal places as it allows readability of the table, also in view of the request of Reviewer 2 to provide 25th and 75th percentile.

Discussion

Rows 199-200: I do not understand what you mean by strictly integrated.
Authors: we replaced “strictly” with “vertically”. It means that the company controls all production stages. 

Rows 207-209: I do not understand this sentence.
Authors: we have rephrased the sentence

Rows 217-219: Neither do I understand this sentence.
Authors: we have rephrased the sentence

Row 232: Check the spelling of Brachyspira hyodysenteriae
Authors: addressed accordingly

Row 238: I do not think that justified is the appropriate word. Perhaps explain would fit better.
Authors: addressed accordingly

Row 243-247: to be able to discuss along these lines you need to provide more detailed information on the differences between Italian welfare friendly and conventional systems. The situation may be completely different in other countries.
Authors: we have added more detailed information. Rows 239-246

Rows 249-250: I think that the work performed to increase the number of farms rearing undocked pigs is commendable as strictly speaking, routine docking is not legal within the EU although in most, but not all countries, this is routine practice.
Authors: thanks for the nice comment on Italian effort

Conclusions

You cannot draw conclusion for the entire Italian pig sector based on the results for only 36 herds.
Authors: we rephrased accordingly

Round 2

Reviewer 1 Report

Dear authors,

thank you for addressing my comments and revising the manuscript. I am happy to recommend that it should be accepted for publication.

Reviewer 2 Report

General comments

The manuscript has improved considerably after the revision and many issues have been sorted by having the manuscript seen by a fluent English-speaking person. There are however some minor issues which need to be addressed before the paper can be published. I suppose that you can provide examples of the used DDDita through the corresponding author in case of interest in specific details.

Specific comments

Title

I believe that “its” should be replaced with the.

Introduction

Rows 59-60: I do not believe that “reviewed” is the appropriate word for the process mad by EMA. As it is written now it sounds as if EMA reviewed the work by WHO when they instead made another categorization in four classes “according to the potential consequences to public health when used in animals”.

Materials and Methods

Rows 80-81: It is still not clear exactly what is implied with “unit”. I take it is fattening places, but it could also mean an entire building. Please clarify.  

Row 84: Who visited the farms and how many visits were made (see row 89)?

Rows 109-112: Was the number of slaughtered pigs used in the equation the number produced for each of the herds enrolled in the study?

Results

Table 1: In the caption, you still write PCU which is misleading. I believe you can write “Defined daily doses per animal” as in the caption for table 2.

Table 2: It is not clear what is implied with “less or more than 2000 pigs” Is it per batch? If this is the case, this should be written out in full to clarify. Furthermore, “other” seems very unspecific.

Discussion

Rows 194-195: As you have changed the title, I think that this sentence must be revised. Begin with the AMU and end the sentence with husbandry practices to be in agreement with the title.

Rows 195-196: Still, this sentence does not make sense to me. Should it be read as that this method allowed for adding different active substances together to make a sum?

Rows 197-198: The second part of the sentence is difficult to read and understand. Do you mean something like this: “so that a reduction could not be detected”?

Rows 198-202: This is an awfully long sentence and its content is difficult to understand. It should be broken down into to or more sentences.

Rows 205-207: Were the percentages for the three classes really the same over all three years?

Row 210: Replace “molecules” with substances.

Rows 210-213: Are these the most common infections which are treated also in Italy? The references describe the situation in other countries such as Korea.

Row 216: Replace “evidenced” with observed.

Rows 216-217: The second half of the sentence is difficult to understand. Please rephrase.

Row 223: Does “AMR levels” refer to those in general, humans or animals? Not clear as written now.

Row 256: The threshold used for ammonia is relatively high, could this have influenced the result that there were not any differences in AMU? Perhaps a lower threshold could have resulted in significant differences. Please explain why this threshold was chosen.

Conclusions

Row 266: Replace managerial with management.

Row 267: I do not understand the second half of the sentence. There are double negations.